# Financing Benefits and Barriers to Routine HIV Screening in Clinical Settings in the United States: A Scoping Review

**DOI:** 10.3390/ijerph20010457

**Published:** 2022-12-27

**Authors:** Hani Serag, Isabel Clark, Cherith Naig, David Lakey, Yordanos M. Tiruneh

**Affiliations:** 1Department of International Medicine, School of Medicine, University of Texas Medical Branch (UTMB), Galveston, TX 77555, USA; 2HIV/STD Prevention & Care Unit, Texas Department of State Health Services, Austin, TX 78714, USA; 3MPH Program, School of Public and Population Health, University of Texas Medical Branch (UTMB), Galveston, TX 77555, USA; 4Administration Division, University of Texas System, Austin, TX 78701, USA; 5Department of Preventive Medicine and Population Health, School of Medicine, University of Texas Tyler, Tyler, TX 75799, USA; 6Department of Internal Medicine, University of Texas Southwestern Medical Center, Dallas, TX 75390, USA

**Keywords:** HIV, routine screening, HIV prevention, financial benefits, opt-out approach

## Abstract

The Centers for Disease Control and Prevention recommends everyone between 13–64 years be tested for HIV at least once as a routine procedure. Routine HIV screening is reimbursable by Medicare, Medicaid, expanded Medicaid, and most commercial insurance plans. Yet, scaling-up HIV routine screening remains a challenge. We conducted a scoping review for studies on financial benefits and barriers associated with HIV screening in clinical settings in the U.S. to inform an evidence-based strategy to scale-up routine HIV screening. We searched Ovid MEDLINE^®^, Cochrane, and Scopus for studies published between 2006–2020 in English. The search identified 383 Citations; we screened 220 and excluded 163 (outside the time limit, irrelevant, or outside the U.S.). Of the 220 screened articles, we included 35 and disqualified 155 (did not meet the eligibility criteria). We organized eligible articles under two themes: financial benefits/barriers of routine HIV screening in healthcare settings (9 articles); and Cost-effectiveness of routine screening in healthcare settings (26 articles). The review concluded drawing recommendations in three areas: (1) Finance: Incentivize healthcare providers/systems for implementing HIV routine screening and/or separate its reimbursement from bundle payments; (2) Personnel: Encourage nurse-initiated HIV screening programs in primary care settings and educate providers on CDC recommendations; and (3) Approach: Use opt-out approach.

## 1. Introduction

In the United States, an estimate of 1.19 million people aged 13 years or older were living with HIV at the end of 2019, of whom approximately 158,500 (13.3%) were undiagnosed [1]. This made the prevalence of HIV among individuals 13 years of age and older 431.5 per 100,000. About 40% of newly diagnosed HIV infections have been transmitted by those living with undiagnosed HIV infections [1]. The high transmission rate from undiagnosed people living with HIV illustrates the importance of routine HIV screening as a primary and secondary preventive measure [2]. The National HIV AIDS Strategy (NHAS) Goals to end the HIV epidemic (2021–2025) include preventing new HIV infections by ensuring healthcare providers are knowledgeable about HIV and prevention recommendations and increasing all persons’ knowledge of their HIV status by implementing the Centers for Disease Control and Prevention (CDC)/United States Preventive Services Task Force (USPSTF) HIV testing recommendations, i.e., universal routine opt-out HIV testing. [2]. Nevertheless, the adoption of routine HIV testing in primary care settings has been limited due to policy, provider, patient, and practice-related factors [3].

The 2006 CDC revised HIV testing recommendations in healthcare settings were released to address the missed opportunities to diagnose persons solely based on risk and to identify persons earlier in the disease process to prevent progression to AIDS-defining conditions and new HIV transmissions. Early diagnosis of HIV, when coupled with early initiation of antiretroviral therapy (ART), can prevent transmission of new HIV infections. Persons with HIV who engage in ongoing HIV medical care and treatment and achieve and maintain an undetectable viral load cannot sexually transmit the virus to others (Undetectable = Untransmissible) [4,5]. ART also reduces susceptibility to opportunistic infections and associated comorbidities. Moreover, routine HIV screening makes HIV testing more equitable with respect to demographic characteristics, socioeconomic factors, and insurance status. Ultimately, universal implementation of screening should further normalize testing for HIV and reduce stigma and bias among patients and providers [6]. For those diagnosed as HIV-negative, routine HIV testing is an opportunity to provide counseling to educate persons about the behaviors that put them at increased risk of acquiring HIV and how to avoid the risks, while those tested positive receive counseling as a part of the linkage to care [5,7]. HIV routine screening and rapid linkage to care should be recognized as two of several preventive measures that include Pre-Exposure Prophylaxis (PrEP), routine monitoring of viral load among those on antiretroviral drugs, and monitoring and addressing HIV drug resistance.

The CDC recommends that everyone between 13 and 64 years of age be tested for HIV at least once as a routine healthcare procedure. Individuals presenting a higher risk should be tested at least once a year [1]. In 2013 the USPSTF endorsed screening for HIV infection among all adolescents and adults aged 15 to 65 years as one of its Grade A recommendations; younger adolescents and older adults at increased risk of infection should also be screened. All pregnant women, including those present in labor or at delivery whose HIV status is unknown, should be screened [8]. In response to the 2013 USPSTF Grade A recommendations, the Center for Medicare & Medicaid Services (CMS) expanded screening coverage, including annual voluntary screening for beneficiaries aged 15 to 65 years, without regard to an individual’s perceived risk. Coverage for pregnant women extends to a maximum of three voluntary HIV screenings. However, despite the CMS expanded coverage, the adoption of routine HIV screening using the opt-out approach, where a patient is informed that HIV testing will be performed unless the patient declines testing, remains operational in clinical settings on a relatively small scale [9]. This might be attributed to the lack of direct financial benefits for healthcare systems to perform routine screening. Unlike the cost-effectiveness of routine HIV screening, the financial benefits and barriers to routine screening in clinical settings are understudied. A better understanding of potential benefits (e.g., cost-saving intervention as knowing the HIV status and rapid linkage to care reduces avoidable hospitalization) and barriers (e.g., lack of unified reimbursement schemes) should result in recommendations to make routine HIV screening more beneficiary to healthcare systems.

We reviewed the current literature that addresses the financial benefits and barriers associated with routine HIV screening in clinical settings and synthesized the arguments we found to inform evidence-driven policy proposals that could contribute to scaling up routine HIV screening. It aims specifically to (1) identify the financial benefits and barriers associated with the implementation of HIV screening in clinical settings within the United States; and (2) identify and describe best practices for maximizing the financial benefits of and overcoming the financial barriers associated with the implementation of HIV screening in clinical settings within the United States.

## 2. Materials and Methods

### 2.1. Sources and Eligibility Criteria

We used Arksey and O’Malley’s framework [10] to conduct a scoping review for studies of the financial benefits and barriers associated with HIV screening in clinical settings in the United States by searching Ovid MEDLINE^®^, Cochrane, and Scopus for studies published between January 2006 and the end of August 2020. We limited the search to articles published in English.

### 2.2. Search Strategy

We used the following Medical Subject Headings (MeSH) terms:HIV: HIV or human immunodeficiency virus, AND;Screening: (screen OR screening) OR (test OR testing OR HIV test), AND;United States: (including all states and excluding all other countries);Financial benefits and financial barriers: (cost benefit* or cost effective* or cost utility or (economic adj2 evaluat*) or marginal analys* or financial benefit* or financial barrier* or (3acilit* adj2 (barrier* or benefit* or 3acility3*))) AND;Clinical settings: (emergency room* or emergency department* or emergency service*) OR (inpatient* or outpatient* or hospital* or ambulatory care or ambulatory clinic* or community health center* or healthcare faciltit* or health care acility*or or health 4acility*).

The search was made by an experienced medical librarian.

### 2.3. Selection and Data Charting

Each citation resulted from the search was evaluated independently by two of the research team by applying the eligibility criteria. Discrepancies were thoroughly discussed by the team to make a collective decision to included and excluded citations. Then included citations were distributed to the research team to extract relevant data and develop a synthesis using the following data items.

### 2.4. Data Items

We categorized eligible studies/articles using the following topics or themes as criteria:Financial benefits of, and barriers to, HIV screening in healthcare settings;Cost-effectiveness of routine HIV screening in clinical settings. Some refer to this as mass/general. We will use the term routine for the remainder of the paper.

The categorization was made during the selection process. Each included citation was categorized by two members of the research team independently then discrepancies were addressed by the whole team collectively.

## 3. Results

The search identified 383 Citations; we screened 220 and excluded 163 (outside the time limit, irrelevant, or outside the U.S.). Of the 220 screened articles, we included 35 and disqualified 155 (did not meet the eligibility criteria)—refer to Figure 1.

We organized the results of this review using the themes used as criteria for ranking the eligible articles.

### 3.1. Financial Benefits of and Barriers to HIV Screening in Healthcare Settings

Nine studies/articles discussed the benefits and barriers associated with general HIV screening in clinical settings [11,12,13,14,15,16,17,18,19]. One study found financial benefits in terms of cost savings, and eight studies/articles discussed the increased costs associated with expanding HIV screening and provided recommendations for reducing those costs.

Li et al. [11] conducted a cost-utility analysis to evaluate the use of point-of-care (POC) rapid HIV testing in three settings in Rhode Island: healthcare facilities, community-based organizations, and partner notification service programs. The study concluded that HIV testing was cost-saving (with a cost-utility of 47,667) in healthcare/clinical settings and was cost-effective in both community settings (community-based organizations) and partner notification service programs (with cost-utilities of USD12,959 and USD14,725, respectively). The cost-saving and cost-effectiveness benefits were consistent across the three settings despite variations in testing costs per positive case and positivity rates. The cost-utility analysis is commonly used when we focus on the quality rather than the length of life. The study used the standard interpretation of cost-utility: cost-saving if cost-utility (R) < USD0 per quality-adjusted life year (QALY); cost-effective if R < USD100,000 per QALY; not cost-effective if R > USD100,000 per QALY [11].

Five studies calculated the costs associated with routine HIV screening in various clinical settings using multiple testing technologies, including POC rapid antibody testing of blood or oral fluid specimens and lab-based antibody/antigen testing. They all concluded that the cost of routine testing in healthcare settings was relatively low and within expected budgets across technologies [12,13,14,15,16]. Gidwani et al. [12] conducted a budget impact analysis to assess the cost associated with non-targeted HIV screening using POC rapid testing. Their results showed that non-targeted testing does not substantially increase the cost compared to the diagnostic testing approach (driven by symptoms associated with an individual’s medical history). The study adopted assumptions of 1% HIV prevalence and 80% test acceptance; the cost of POC was USD1,418,088, vs. USD1,320,338 for Usual Care (*p* = 0.5854). Spaulding et al. [14] concluded that routine HIV testing in emergency departments and jails, using HIV seropositivity tests, resulted in new diagnoses at a cost that falls within a range comparable to costs cited in published reports.

Schackman et al. [15] studied the cost associated with expanded HIV testing from POC test kits to rapid-result laboratory testing for patients having blood drawn for clinical reasons in emergency departments in the Bronx, New York, and Washington, DC. The results showed that expanded testing was associated with a general increase in expenditures and a heavier financial burden. However, hypothesized analysis of HIV testing with automated steps using electronic medical records showed a 45% cost reduction for non-reactive tests (the vast majority of performed tests) and a 20% reduction for reactive tests. Anaya et al. [16] conducted a facility-specific budget impact analysis of expanded HIV testing and care in the Veterans Affairs healthcare system. The analysis showed that expanded HIV testing was associated with an increase in newly identified HIV-positive cases, and the greater budget impact was associated with the provision of antiretroviral therapy (care) rather than expanded HIV testing. The study also concluded that the budgetary implications of expanded testing were greater during the initiation stage and declined gradually thereafter.

Sison et al. [17] used a qualitative approach to understand the attitudes of local providers in the Mississippi Delta region regarding HIV testing and care. They conducted 25 in-depth interviews with primary care providers and infectious disease specialists. Among the many other results reported, the study identified reimbursement structures, in particular payment schemes known as bundled payments, as a barrier to adopting routine HIV screening in healthcare settings [17]. Bundled payments reimburse healthcare providers (hospitals or physicians) on the basis of defined clinical episodes. This means that providers or healthcare-providing organizations receive the same reimbursement for a certain pre-defined type of case regardless of the requested examinations or procedures performed [20]. Accordingly, providers and healthcare-providing organizations may not receive additional reimbursement for the HIV test. Bartlett et al. [18] reviewed legal, financial, and organizational challenges associated with implementing the CDC recommendation of routine HIV screening. Although this is not unique to HIV, variability in payments as well as in payment structures based on insurance status/insurance coverage and location was identified as one of the challenges in adopting routine screening.

Mehta et al. [19] used an anonymous online survey to assess knowledge of guidelines, practices, and perceived barriers to HIV screening of adolescents in emergency departments. The results show that nearly two-thirds of participating providers perceived cost-ineffectiveness as a barrier. However, most studies that have conducted cost-effectiveness or cost-savings analyses at considerably low thresholds have reported favorable results for routine screening [19]. The low thresholds reported in this report concluded that, methodologically, these studies included low financial rewards, in terms of reimbursement or direct payments, in their calculations of the associated health gains. The study also suggested that providers’ knowledge or perceptions of the cost-effectiveness of HIV screening is a barrier that needs to be addressed, perhaps through continuing education programs [19].

In summary, these studies indicate that (1) routine HIV testing is a financially feasible and generally cost-saving intervention in healthcare settings, and (2) variability across current reimbursement structures and provider misperceptions of the cost-effectiveness of routine screening is a barrier to adoption.

### 3.2. Cost-Effectiveness of Routine HIV Screening in Clinical Settings

Five studies have examined the cost-effectiveness of routine HIV screening in healthcare settings, including inpatient, outpatient, and emergency departments [21,22,23,24,25], and twenty-one studies have examined either the cost-effectiveness, cost savings, or cost benefits of implementing HIV routine screening for specific population/patients groups such as antenatal mothers and criminal justice-involved individuals (CJI) as summarized below [26,27,28,29,30,31,32,33,34,35,36,37,38,39,40,41,42,43,44,45,46,47,48,49,50,51,52,53,54,55,56].

Two studies used mathematical cost-effectiveness models in their analyses. Farnham et al. [21] developed a mathematical model to evaluate the cost of every new HIV diagnosis per quality-adjusted life-year in screening programs in clinical settings. The study concluded that routine HIV screening is cost-effective across a wide range of testing regimes and positivity rates. Sanders et al. [22] conducted a cost-effectiveness analysis using a Markov model [57] to examine the costs and benefits of strategies for improving HIV testing. The data were driven by a controlled randomized trial. The study assessed cost-effectiveness by type of provider and testing technology. The findings indicated that using routine rapid testing in nurse-initiated programs in primary care settings was more cost-effective than testing under other counseling-based models that were examined [22].

Owens et al. [23] used multivariate analysis to identify and assess the prevalence of HIV in inpatient and outpatient settings at six healthcare sites. They concluded that the prevalence of new HIV diagnoses was high enough (above 0.1%) to exceed the standard threshold of cost-effectiveness. The standard cost-effectiveness threshold is usually related to benchmark interventions. In the United States, the threshold of cost-effectiveness that is commonly used for HIV screening is USD100,000 for each gained quality-adjusted life year (QALY) [58].

Torres et al. [24] used a self-administered questionnaire to assess the reported cost-effectiveness of routine HIV screening in six emergency departments. The selection of the participating emergency departments was made to achieve geographic diversity (South, Midwest, West and Northeast regions of the nation), type (public, nonprofit, and private), size (20–68 beds), HIV screening model (targeted, non-targeted, and universal), and testing process (point-of-care, state laboratory, and hospital laboratory). Despite variations in the structure and process across the participating emergency departments, there was general agreement that HIV screening programs are cost-effective according to standard thresholds defined by the authors that align with thresholds used in several similar studies [24].

In their randomized controlled trial, Wagner et al. [25] assessed the effectiveness of two interventions: (1) an opt-out approach and (2) offering financial incentives to patients to opt in to HIV testing. The study was conducted for one year in an emergency department and used a POC rapid test, making results available in 1–2 h. The study concluded that providing patients with financial incentives and using the opt-out approach for HIV screening were superior interventions compared to combining no financial incentives with the opt-in approach. The study also suggested that opt-out HIV screening coupled with financial incentives was more cost-effective [25].

Two studies focused on the cost-effectiveness of HIV screening among women during the perinatal period, primarily in the late gestational and antenatal period [36,37]. In their systematic review, Bert et al. [36] concluded that universal antenatal HIV screening and rescreening in late pregnancy are cost-effective in both developing and developed countries (including the United States). The review also found higher cost-effectiveness in settings with higher HIV burdens. An earlier systematic review by Ibekwe et al. [37] reported similar results, indicating that the opt-out approach is cost-effective when conducting routine antenatal HIV screening. The review also found a strong association between the level of cost-effectiveness and HIV prevalence in the setting under study; the cost-effectiveness of screening is greater in settings with higher HIV prevalence.

Scott et al. [46] conducted a cost-effectiveness analysis to compare two opt-out HIV testing approaches during delivery that was limited to mothers who (1) had not undergone previous third-trimester screening or (2) had not undergone prenatal screening. They concluded that universal HIV screening during delivery is cost-effective even in areas with annual cumulative HIV incidence rates of <0.02% for reproductive-age women. Such screening reduces the rate of mother-to-child transmission. In these areas, the cost-effectiveness benchmark would be USD89,926.94 per QALY, which is close to the commonly considered break-even threshold of USD100,000 per QALY.

Cipriano et al. [38] indicated that HIV screening among illicit drug users is modestly cost-effective. They argued, however, that more frequent HIV screening of drug users adds additional benefits at a lower cost. Their conclusions also apply to screening for hepatitis C virus (HCV) infection. In their systematic review, Harawa et al. [39] indicated that screening for HIV and sexually transmitted diseases (STDs) in CJI populations is cost-efficient and may help to reduce HIV transmission between Black MSM as a subset of this population.

Two studies examined the cost-effectiveness of HIV screening in multiple age groups [40,41]. Neilan et al. [40] used a simulation model to assess the optimal age for one-time HIV screening. Their results suggest that one-time routine HIV screening at age 25 may produce the best clinical outcomes while being cost-effective. Sanders et al. [41] applied incremental cost-effectiveness analysis. The study concluded that the cost-effectiveness of routine HIV screening for people aged 55–75 years is controversial, even if the screening is performed in a population with an HIV prevalence of 0.1% or greater.

Stevens et al. [42] used a computer simulation model to assess the cost-effectiveness of three population-based HIV screening approaches in a highly populated HIV high-risk urban area (New York City). They concluded that respondent-driven sampling (RDS) with anonymous HIV testing (RDS-A) was more cost-effective than either RDS with a 2-session confidential HIV-testing approach (RDS-C) or venue-based sampling (VBS).

Three studies in our review evaluated the cost-effectiveness of screening for HIV while performing other healthcare procedures, e.g., substance abuse treatment and elective surgeries [44,45,46]. Using a randomized trial, Schackman et al. [44] found that rapid on-site POC HIV antibody testing during substance abuse treatment is cost-effective and increases life expectancy. Two studies investigated the cost-effectiveness of HIV screening during elective surgeries. Nussbaum et al. [45] concluded that routine screening for HCV, but not HIV, during elective cranial neurosurgery is cost-effective. The study sample of 1461 patients who underwent elective craniotomy between July 2009 and July 2016 showed that no patients were living with HIV. In another report, Dowdy et al. [47] suggested that pre-operative screening for blood-borne infections, including HIV, is cost-effective before elective arthroplasty.

The results of nine studies indicate that HIV routine screening in both healthcare and community settings is generally cost-effective, especially when the prevalence is moderately high, and that it is an effective HIV preventive measure [48,49,50,51,52,53,54,55,56]. Eggman et al. [13] also indicated that conducting rapid POC HIV universal screening in STD clinics incurs a relatively low cost.

## 4. Discussion and Recommendations

This scoping review provides compelling evidence that opt-out routine HIV testing is cost-effective in detecting HIV infections and is accompanied by financial gains. The studies we reviewed consistently conclude that routine HIV screening in healthcare facilities can effectively overcome financial obstacles, resulting in widespread facilitation of care and improved healthcare outcomes. It is worth noting the 2006 CDC revised HIV testing recommendations in health care settings were based on the CDC perinatal HIV testing recommendations to prevent vertical transmission to newborns [https://www.cdc.gov/hiv/policies/law/states/perinatal.html] (accessed on 15 October 2022). These recommendations have been proven to be not only successful but also financially feasible as evidenced by a majority of states in the United States have adopted laws to ensure all pregnant women are tested for HIV during the 3rd trimester, during labor and delivery when HIV status is not documented, and of the newborn if the mother’s HIV status remains unknown.

We found that the implementation of routine HIV screening is affected by variability in reimbursement structures and provider financial benefits. Reimbursement structures that vary based on insurance status, insurance coverage, the payer, and location pose challenges [18]. Variations in payment and reimbursement structures and low financial rewards for healthcare providers when conducting universal HIV testing also represent a major concern [17,18,19]. These variations reflect a multiplicity of policies and payers and the nature of contractual relationships between payers, intermediating contractors, and healthcare providers. In 2018, 35.1% of healthcare payments were tied to bundled payment models [59]. Bundled payments prevent healthcare systems from reaping the direct financial benefits of adopting routine HIV screening in hospital inpatient, outpatient, and emergency departments. While instituting bundled payments is a step toward value-based payments, which gives more weight to patient outcomes [59], it does not reimburse or incentivize healthcare providers for activities or procedures that are not closely related to cases that include HIV screening or other preventive measures [60]. The high rates of un- or underinsured patients that emergency departments serve is also one of the barriers to implementing prevention measures, including HIV testing programs.

Routine testing is cost-effective and can be financially attractive with a unified payment structure that is separated from any bundled payments. This will make HIV screening financially beneficial to healthcare systems regardless of the insurance coverage or modality. A new temporary financial mechanism to cover the cost of testing for the uninsured may also enhance the adoption of routine HIV testing in clinical settings. Additionally, routine HIV testing was designated by the USPSTF as a Grade A preventive service in 2013; the addition of the modifier 33 to the CPT code alerts payers that this is a preventive service that is required to be provided to patients and covered at no cost to the patient. It is imperative to emphasize that the main budget impact for HIV is associated with the treatment of people living with HIV and not the testing [16].

Our review found that the opt-out HIV screening approach and nurse-initiated HIV screening programs are more effective than their counterparts [9,18,37,61,62,63,64,65,66,67,68,69] and suggests that nurse-initiated HIV screening programs in primary care settings are more cost-effective than other types of screening programs [22]. This conclusion is consistent with the results reported in other studies. Henry et al. [70] have argued that HIV testing of the general population is generally cost-effective, but nurse-initiated HIV screening programs streamline HIV testing, education, and counseling for persons diagnosed with HIV. Spaulding et al. [14] also have associated the use of nursing staff with low-cost screening in emergency departments and jail settings.

This review presents strong evidence that using an opt-out versus an opt-in approach to HIV testing in clinical settings facilitates the identification of HIV in undiagnosed persons unaware of their HIV status. Routine screening addresses these missed opportunities to diagnose patients and to refer them to HIV medical treatment and other support services to achieve and maintain viral suppression eliminating the potential of new HIV transmissions. Additionally, routine screening eliminates the stigma and bias associated with HIV. If implemented according to the recommendations, screening can be offered to adolescents before they become sexually active with the opportunity to educate them about the risks associated with acquiring HIV and how to prevent or avoid them. Persons with HIV also live longer, and many who were diagnosed during the early days of HIV/AIDS may have fallen out of care. Screening all patients regardless of risk or advanced age, i.e., ages 55 to 75, which was previously identified as controversial [41], can also present the opportunity to identify patients who do not disclose their HIV status and give them another chance to re-engage in HIV care to achieve and maintain viral suppression. Finally, using the opt-out approach in HIV screening was not associated with any significant decline in patient satisfaction or reporting of risk behavior [67,69].

Financial incentives to patients were found to contribute to the increase in testing acceptability [25]. However, the sustainability of financial incentives to patients is questionable. In their randomized clinical trial, Montoy, et al. [71] concluded that the use of opt-out HIV screening increased patient testing acceptability by 23.9% compared to an increase in acceptability of 1%, 10.5%, and 15.0% when offered cash incentives of USD1, USD5, and USD10, respectively.

Although research supports the cost-effectiveness of routine HIV screening in healthcare settings, the lack of knowledge among providers of recommended HIV prevention practices and its cost-effectiveness is a barrier to implementing routine HIV screening programs [19,60]. Recommendations to overcome these barriers include integrating HIV prevention training in all healthcare professional school curriculums, implementing tagged continuing education opportunities about routine HIV prevention and screening, especially for those providing primary and emergency care.

One shortcoming of this review was the lack of clear identification of test technology employed in many of the articles reviewed. When considering the cost-effectiveness of routine HIV screening, it is important to identify the best test technology to implement based on the type of healthcare setting, specifically primary or emergency care. Both POC test kits and lab-based testing in accordance with the CDC/APHL HIV Testing Algorithm are effective tools for diagnosing HIV. When choosing a testing technology, considerations should include the time required of staff to collect the specimen and the time and efficiency to perform, interpret and document the test result. Although POC test kits offer rapid results within one to twenty minutes, they also require hands-on time by staff and may be subject to human error when interpreting and documenting test results. Additionally, a positive POC test must be confirmed with a lab-based conventional blood draw according to the HIV testing algorithm. POC test kits, however, may be more effective in the primary care setting to enable immediate preliminary results delivery, increasing the chance the patient will return for confirmatory results and referral to care.

Emergency departments are set up for conventional lab-based testing on random-access analyzers that can produce preliminary results in 30 min to an hour. Cost-effective benefits include that the HIV test can be performed on the same serum specimen submitted for other tests ordered and processed in real-time in the hospital E.D. lab, with expedited results documented automatically in the electronic health record system. Compared to POC test technology, lab-based testing can detect the HIV p24 antigen, which identifies acute or early infection before the presence of antibodies is detected, offering the patient the benefit of access to rapid ART treatment long before the POC test detect HIV.

Early oral signs of HIV qualify dental settings to be optimal places for early detection. The adoption of HIV routine screening in dental facilities may significantly expand the coverage of HIV testing [72]. We also recommend benefiting from the experiences of other countries in the world, which needs further research work.

HIV routine screening is a major step towards ending the HIV endemic in the US, which will save the nation more than twenty-eight billion US Dollars as federal spending on HIV care and treatment per year (based on the estimates of 2022) [73].

## 5. Conclusions

We note overall that routine HIV testing is financially feasible and generally cost-saving. Nevertheless, reimbursement varies in structure, and providers often fail to understand the cost implications, discouraging the adoption of routine screening. The effect of undiagnosed HIV on healthcare costs was not included in this review, but there are many anecdotal cases of multiple healthcare visits to primary care and emergency departments due to symptoms related to HIV; diagnosing HIV in these patients provides the opportunity to refer them to HIV medical care and treatment to achieve viral suppression and eliminate many of unnecessary healthcare costs associated with undiagnosed HIV. Implementing routine HIV screening in healthcare settings will go a long way to contributing to the National HIV/AIDS Strategy goals to end the HIV epidemic in the United States by 2030.

## Figures and Tables

**Figure 1 ijerph-20-00457-f001:**
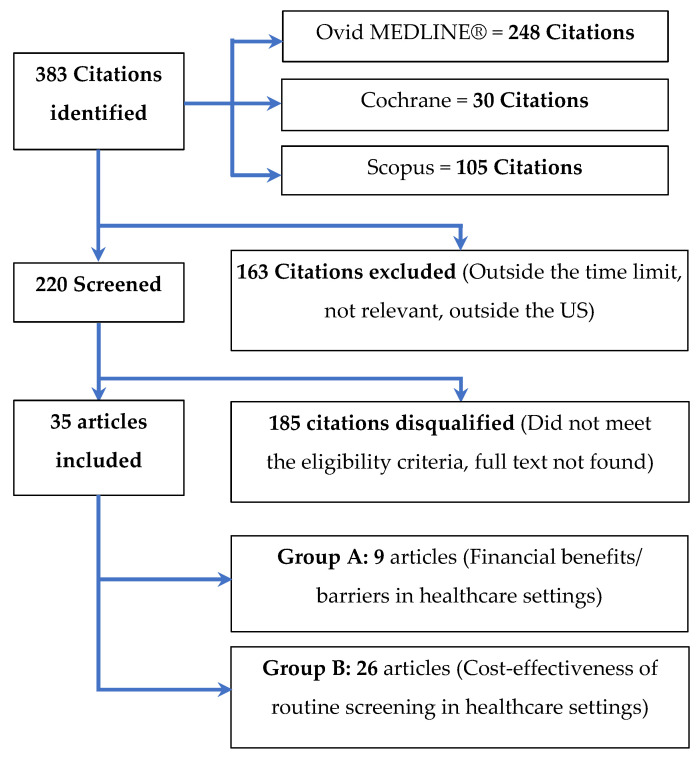
The selection process for articles included in the scoping review.

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
