# Peer review of "Financing Benefits and Barriers to Routine HIV Screening in Clinical Settings in the United States: A Scoping Review"

_ijerph, 2022, doi:10.3390/ijerph20010457_

Round 1

Reviewer 1 Report

Dear Authors,

The study titled “Financing Benefits and Barriers to Routine HIV Screening in Clinical Settings: A Scoping review” covered the subject in detail and presented it appropriately. It was seen that the limited aspects of the study were discussed in the discussion section. The English of the article is at an acceptable level. I recommend that this work be accepted.

Best regards

Author Response

Comment (1)

The study titled “Financing Benefits and Barriers to Routine HIV Screening in Clinical Settings: A Scoping review” covered the subject in detail and presented it appropriately. It was seen that the limited aspects of the study were discussed in the discussion section. The English of the article is at an acceptable level. I recommend that this work be accepted.

Response (1)

Thank you very much; we do appreciate your kind comment.

Reviewer 2 Report

Dear authors

congrats for the work done

I analysed the paper and I make the following observations:

In general, it would be useful to correct some typos.

Title: add the name of the location of the study

Methods: insert figure 1 in materials and methods

Results:

- in line 277 write figure and table but there isn't table but only a figure;

-  too much extensive, reduce

Discussions: too much extensive, reduce.

Author Response

Comment (1)

In general, it would be useful to correct some typos.

Response (1)

Thank you. We conducted language editing and ensured that typos were corrected.

Comment (2)

Title: add the name of the location of the study

Response (2)

Thanks. We added “in the United States” The tile now reads “Financing Benefits and Barriers to Routine HIV Screening in Clinical Settings in the United States: A Scoping review.”

Comment (3)

Methods: insert figure 1 in materials and methods

Response (3)

Thanks. There is a certain place in the template for tables and figures. So, we will leave this to the journal to address. We agree that the right place would be in the materials and methods.

Comment (4)

Results:

  • in line 277 write figure and table but there isn't table but only a figure
  • too much extensive, reduce

Response (4)

Thanks. This was an instruction for the template use that we forgot to remove. Now removed.

We have revised it and shortened it as much as we could after addressing the comments of other reviewers.

Comment (5)

Discussions: too much extensive, reduce.

Response (5)

We have revised it and shortened it as much as we could after addressing the comments of other reviewers.

Reviewer 3 Report

Financing Benefits and Barriers to Routine HIV Screening in Clinical Settings: A Scoping review 

Line 30 and 31: include the citation/reference for this sentences

Line 47: Early diagnosis of HIV, when coupled with early initiation of antiretroviral therapy (ART), can prevent transmission of new HIV infections. Authors needs to also include HIV drug resistance (HIVDR) and routine viral load monitoring as other important in HIV transmission and prevention. 

Line 55: For those diagnosed as HIV-negative, routine HIV testing is an opportunity to provide counseling to educate persons about the behaviors that put them at increased risk of acquiring HIV and how to avoid the risks [5,7]. It is important to do the same counseling for those who test positive.

Line 59: The CDC recommends that everyone between 13 and 64 years of age be tested for HIV at least once as a routine healthcare procedure. Is this at least once a year or life time?

Line 230: Can authors quantify in USD the difference in the level of cost-effectiveness and HIV prevalence in higher and low-income settings or lower and high HIV prevalence settings. This is particularly of great importance for planning purposes for governments and insurance providers.

Line 241: Should be Illicit drug users

Generally under the discussion: Authors should discuss the  trends of HIV self-test (or rapid self-test) in the contest of their findings as well as the average cost of HIV treatment per year for an individual living with HIV (this includes the viral loading monitoring and HIVDR testing).

Author Response

Comment (1)

Line 30 and 31: include the citation/reference for this sentences

Response (1)

Thanks, we added the reference.

Comment (2)

Line 47: Early diagnosis of HIV, when coupled with early initiation of antiretroviral therapy (ART), can prevent transmission of new HIV infections. Authors needs to also include HIV drug resistance (HIVDR) and routine viral load monitoring as other important in HIV transmission and prevention.

Response (2)

Thanks. We have added a short sentence at the end of the paragraph to address this comment. We made a concise addition to keep the focus on the routine screening. “HIV routine screening and rapid linkage to care should be recognized as two of several preventive measures that include Pre-Exposure Prophylaxis (PrEP) for people with increased risk of infection as well as routine monitoring of viral load among those on antiretroviral drugs, and monitoring and addressing HIV drug resistance.”

Comment (3)

Line 55: For those diagnosed as HIV-negative, routine HIV testing is an opportunity to provide counseling to educate persons about the behaviors that put them at increased risk of acquiring HIV and how to avoid the risks [5,7]. It is important to do the same counseling for those who test positive.

Response (3)

Thanks, we added a short sentence to address your comment. “ . . . while those tested positive receive counseling as a part of the linkage to care.”

Comment (4)

Line 59: The CDC recommends that everyone between 13 and 64 years of age be tested for HIV at least once as a routine healthcare procedure. Is this at least once a year or life time?

Response (4)

CDC recommends that everyone between the ages of 13 and 64 get tested for HIV at least once as part of routine health care. So, it is once in life. However, people who may be at high risk for HIV should be screened at least annually. We did not address this point about the people at high risk since we are focusing on routine screening.

Comment (5)

Line 230: Can authors quantify in USD the difference in the level of cost-effectiveness and HIV prevalence in higher and low-income settings or lower and high HIV prevalence settings. This is particularly of great importance for planning purposes for governments and insurance providers.

Response (5)

This is a great suggestion that would support policy proposals. However, we think that the scope and methods of this review would not allow doing so. This will need a rigorous new cost-effectiveness study or meta-analysis of existing cost-effectiveness studies. 

Comment (6)

Line 241: Should be Illicit drug users

Response (6)

Thanks. We added “illicit.”

Comment (7)

Generally under the discussion: Authors should discuss the trends of HIV self-test (or rapid self-test) in the contest of their findings as well as the average cost of HIV treatment per year for an individual living with HIV (this includes the viral loading monitoring and HIVDR testing).

Response (7)

Thanks, these are great suggestions. However, it might need a different article focusing on the cost-effectiveness of different methods of HIV testing. The current focus of this review is on the financial benefits and barriers to HIV screening in clinical settings, which will not include self-testing. Yet, we have added a short paragraph at the end to briefly discuss the treatment cost “HIV routine screening is a major step towards ending the HIV endemic in the US, which will save the nation more than twenty billion US Dollars as federal spending on HIV care and treatment per year (based on the estimates of 2019) [74].

Reviewer 4 Report

1. Introduction

The introduction is well written. A detailed description regarding the current HIV prevalence and management guidelines in the US is presented to the readers.

It would be great if the authors can further elaborate Line 69-72 why HIV testing still remains on “a relatively small scale”. The authors may discuss briefly the potential financing benefits and barriers in the introduction to explain the current knowledge gap, leading the readers to understand further why this review is performed.

2. Methodology

The methodology is very brief, many details are lacking. Although the procedures of scoping review are less strict than systematic review, further details of the review should be reported.

I would advise the authors to follow the PRISMA checklist for scoping review or other validated guidelines to revise the reporting of the methodology section (https://prisma-statement.org/documents/PRISMA-ScR-Fillable-Checklist_11Sept2019.pdf)

3. Results

a. The author has summarised the findings and reported their qualitative synthesis. However, the authors should quote data reported from their included studies (mean difference (SD), ICER, p-value, 95% CI etc) to support their findings.

Some examples are listed below

Line 120-122 

Supporting data that HIV testing saved costs (e.g. mean difference (SD), p-value, 95% CI etc)?

Line 131-133

Any p-value to show no significance difference?

Line 142-143

p-value and 95% CI?

Line 146-147

OR or mean (SD)? p-value and 95% CI?

b. Did the author attempt to perform meta-analysis/ qualitative synthesis for the included studies.

e.g. page 3/12 Line 125 can findings from the 5 included studies be pooled for meta-analysis? Or meta-analysis cannot be performed due to heterogeneity of results?

4. Discussion

The authors have thoroughly discussed the findings in the results. However, the scope of the scoping review can be further extended to discuss the following aspects, so as to present a more comprehensive picture of HIV screening to the audience:

a.The early and obvious signs of HIV include oral candidiasis and other orofacial conditions. It would be interesting for the authors to discuss the oral manifestations of individuals infected with HIV and the feasibility and cost-effectiveness for implementing HIV screening in a dental setting. The authors can refer to the below references for details.

Lam, P. P. Y., Zhou, N., Wong, H. M., & Yiu, C. K. Y. (2022). Oral Health Status of Children and Adolescents Living with HIV Undergoing Antiretroviral Therapy: A Systematic Review and Meta-Analysis. International journal of environmental research and public health, 19(19), 12864. https://doi.org/10.3390/ijerph191912864

Lam, P. P. Y., Chua, H., Ekambaram, M., Lo, E. C. M., & Yiu, C. K. Y. (2022). Does Early Childhood Caries Increase Caries Development among School Children and Adolescents? A Systematic Review and Meta-Analysis. International journal of environmental research and public health, 19(20), 13459. https://doi.org/10.3390/ijerph192013459

Campo, J., Cano, J., del Romero, J., Hernando, V., del Amo, J., & Moreno, S. (2012). Role of the dental surgeon in the early detection of adults with underlying HIV infection/AIDS. Medicina oral, patologia oral y cirugia bucal, 17(3), e401–e408. https://doi.org/10.4317/medoral.17527

Suarez-Durall, P., Osborne, M. S., Enciso, R., Melrose, M. D., & Mulligan, R. (2019). Results of offering oral rapid HIV screening within a dental school clinic. Special care in dentistry : official publication of the American Association of Hospital Dentists, the Academy of Dentistry for the Handicapped, and the American Society for Geriatric Dentistry, 39(2), 188–200. https://doi.org/10.1111/scd.12363

b. Other than just focusing on the findings and practice in the US, the authors should also report and compare their findings and the practice with in other parts of the world.

Guerras JM, Belza MJ, Fuster MJ, Fuente L, García de Olalla P, Palma D, García-Pérez JN, Hoyos J, On Behalf Of The Methysos Project Group. Knowledge and Prior Use of HIV Self-Testing in Madrid and Barcelona among Men Who Have Sex with Men More than One Year after Its Legal Authorization in Spain. Int J Environ Res Public Health. 2022 Jan 19;19(3):1096. doi: 10.3390/ijerph19031096. PMID: 35162118; PMCID: PMC8834423.

Author Response

Comment (1)

Introduction

The introduction is well written. A detailed description regarding the current HIV prevalence and management guidelines in the US is presented to the readers.

Response (1)

Thanks. We do appreciate your comment.

Comment (2)

It would be great if the authors can further elaborate Line 69-72 why HIV testing still remains on “a relatively small scale”. The authors may discuss briefly the potential financing benefits and barriers in the introduction to explain the current knowledge gap, leading the readers to understand further why this review is performed.

Response (2)

Thank you for this comment. We added a few sentences at the end of the paragraph to address your comment. “This might be attributed to the lack of direct financial benefits for healthcare systems to perform routine screening. Unlike the cost-effectiveness of HIV routine screening, the financial benefits and barriers to HIV routine screening in clinical settings are understudied. A better understanding of potential benefits (e.g., cost-saving intervention as knowing the HIV status and rapid linkage to care reduces avoidable hospitalization) and barriers (e.g., lack of unified reimbursement schemes) should result in recommendations to make HIC routine screening more beneficiary to healthcare systems.”

Comment (3)

Methodology

The methodology is very brief, many details are lacking. Although the procedures of scoping review are less strict than systematic review, further details of the review should be reported.

I would advise the authors to follow the PRISMA checklist for scoping review or other validated guidelines to revise the reporting of the methodology section (https://prisma-statement.org/documents/PRISMA-ScR-Fillable-Checklist_11Sept2019.pdf) 

Response (3)

Thank you very much for this comment. We reviewed the methods:

  • We added a heading for the first paragraph, “Sources and eligibility criteria,” and added one sentence at the end of this paragraph, “The search was made by an experienced medical librarian.”
  • We added a new section, “Selection and data charting: Each citation resulted from the search was evaluated independently by two of the research team by applying the eligibility criteria. Discrepancies were thoroughly discussed by the team to make a collective decision on included and excluded citations. Then included citations were distributed to the research team to extract relevant data and develop a synthesis using the following data items.”

We added a heading to the last paragraph, “Data items,” and added an explanatory note at the end of this paragraph “The categorization was made during the selection process. Each included citation was categorized by two members of the research team independently, then discrepancies were addressed by the whole team collectively.”

Comment (4)

Results

a. The author has summarised the findings and reported their qualitative synthesis. However, the authors should quote data reported from their included studies (mean difference (SD), ICER, p-value, 95% CI etc) to support their findings.

Response (4)

Thank you very much for this comment. We have addressed this as indicated below.

Comment (5)

Line 120-122: Supporting data that HIV testing saved costs (e.g. mean difference (SD), p-value, 95% CI etc)?

Response (5)

We have elaborated on the evidence as follows (the added text is underlined).

Li et al. [11] conducted a cost-utility analysis to evaluate the use of point-of-care (POC) rapid HIV testing in three settings in Rhode Island: healthcare facilities, community-based organizations, and partner notification service programs. The study concluded that HIV testing was cost-saving (with a cost-utility of 47,667) in healthcare/clinical settings and was cost-effective in both community settings (community-based organizations) and partner notification service programs (with cost-utilities of $12,959 and $14,725, respectively). The cost-saving and cost-effectiveness benefits were consistent across the three settings despite variations in testing costs per positive case and positivity rates. The cost-utility analysis is commonly used when we focus on the quality rather than the length of life. The study used the standard interpretation of cost-utility: cost-saving if cost-utility (R) <$0 per quality-adjusted life year (QALY); cost-effective if R<$100,000 per QALY; not cost-effective if R>$100,000 per QALY [11].

Comment (6)

Line 131-133: Any p-value to show no significance difference?

Response (6)

We have add the underlined sentence:

Gidwani et al. [12] conducted a budget impact analysis to assess the cost associated with non-targeted HIV screening using POC rapid testing. Their results showed that non-targeted testing does not substantially increase the cost compared to the diagnostic testing approach (driven by symptoms associated with an individual’s medical history). The study adopted assumptions of 1% HIV prevalence and 80% test acceptance; the cost of POC was $1,418,088, vs. $1,320,338 for Usual Care (p = 0.5854).

Comment (7)

Line 142-143: p-value and 95% CI?

Line 146-147: OR or mean (SD)? p-value and 95% CI?

Response (7)

These citations did not provide statistical analysis details

Comment (8)

b. Did the author attempt to perform meta-analysis/ qualitative synthesis for the included studies.

e.g. page 3/12 Line 125 can findings from the 5 included studies be pooled for meta-analysis? Or meta-analysis cannot be performed due to heterogeneity of results?

Response (8)

Yes, the results of 5 studies are significantly heterogeneous, and the methods are different. The meta-analysis was not a possibility to consider.

Comment (9)

Discussion

The authors have thoroughly discussed the findings in the results. However, the scope of the scoping review can be further extended to discuss the following aspects, so as to present a more comprehensive picture of HIV screening to the audience:

Response (9)

Thank you very much.

Comment (10)

The early and obvious signs of HIV include oral candidiasis and other orofacial conditions. It would be interesting for the authors to discuss the oral manifestations of individuals infected with HIV and the feasibility and cost-effectiveness for implementing HIV screening in a dental setting. The authors can refer to the below references for details.

Lam, P. P. Y., Zhou, N., Wong, H. M., & Yiu, C. K. Y. (2022). Oral Health Status of Children and Adolescents Living with HIV Undergoing Antiretroviral Therapy: A Systematic Review and Meta-Analysis. International journal of environmental research and public health, 19(19), 12864. https://doi.org/10.3390/ijerph191912864

Lam, P. P. Y., Chua, H., Ekambaram, M., Lo, E. C. M., & Yiu, C. K. Y. (2022). Does Early Childhood Caries Increase Caries Development among School Children and Adolescents? A Systematic Review and Meta-Analysis. International journal of environmental research and public health, 19(20), 13459. https://doi.org/10.3390/ijerph192013459

Campo, J., Cano, J., del Romero, J., Hernando, V., del Amo, J., & Moreno, S. (2012). Role of the dental surgeon in the early detection of adults with underlying HIV infection/AIDS. Medicina oral, patologia oral y cirugia bucal, 17(3), e401–e408. https://doi.org/10.4317/medoral.17527

Suarez-Durall, P., Osborne, M. S., Enciso, R., Melrose, M. D., & Mulligan, R. (2019). Results of offering oral rapid HIV screening within a dental school clinic. Special care in dentistry: official publication of the American Association of Hospital Dentists, the Academy of Dentistry for the Handicapped, and the American Society for Geriatric Dentistry, 39(2), 188–200. https://doi.org/10.1111/scd.12363

Response (10)

Add a short paragraph

Early oral signs of HIV qualify dental settings to be optimal places for early detection. The adoption of HIV routine screening in dental facilities may significantly expand the coverage of HIV testing [73].

We could not use the other suggested citations, although they are great references due to their focus on children, while our review focus on routine screening for people aged 13 and above according to the SDC recommendation.

Comment (11)

Other than just focusing on the findings and practice in the US, the authors should also report and compare their findings and the practice with in other parts of the world.

Guerras JM, Belza MJ, Fuster MJ, Fuente L, García de Olalla P, Palma D, García-Pérez JN, Hoyos J, On Behalf Of The Methysos Project Group. Knowledge and Prior Use of HIV Self-Testing in Madrid and Barcelona among Men Who Have Sex with Men More than One Year after Its Legal Authorization in Spain. Int J Environ Res Public Health. 2022 Jan 19;19(3):1096. doi: 10.3390/ijerph19031096. PMID: 35162118; PMCID: PMC8834423.

Comment (11)

We added a sentence on the importance of benefiting from the experiences of other countries. We did not expand this part since the focus of this review is on the US using the current model of healthcare financing, which is significantly different from health systems in Europe.

We also recommend benefiting from the experiences of other countries in the world, which needs further research work.

Round 2

Reviewer 4 Report

The authors have answered all my concerns and made appropriate revisions.

I have no further comments.